# Persistence of Multiple Paramyxoviruses in a Closed Captive Colony of Fruit Bats (*Eidolon helvum*)

**DOI:** 10.3390/v13081659

**Published:** 2021-08-20

**Authors:** Louise Gibson, Maria Puig Ribas, James Kemp, Olivier Restif, Richard D. Suu-Ire, James L. N. Wood, Andrew A. Cunningham

**Affiliations:** 1Institute of Zoology, Zoological Society of London, Regent’s Park, London NW1 4RY, UK; mariapuigribas@gmail.com (M.P.R.); jamesrussellkemp@gmail.com (J.K.); 2Royal Veterinary College, University of London, Royal College Street, London NW1 0TU, UK; 3Disease Dynamics Unit, Department of Veterinary Medicine, University of Cambridge, Cambridge CB3 0ES, UK; or226@cam.ac.uk (O.R.); jlnw2@cam.ac.uk (J.L.N.W.); 4School of Veterinary Medicine, College of Basic and Applied Sciences, University of Ghana, Legon, Accra P.O. Box LG 25, Ghana; suuire@gmail.com

**Keywords:** chiroptera, *Pteropodidae*, longitudinal study, *Henipavirus*, *Pararubulavirus*

## Abstract

Bats have been identified as the natural hosts of several emerging zoonotic viruses, including paramyxoviruses, such as Hendra and Nipah viruses, that can cause fatal disease in humans. Recently, African fruit bats with populations that roost in or near urban areas have been shown to harbour a great diversity of paramyxoviruses, posing potential spillover risks to public health. Understanding the circulation of these viruses in their reservoir populations is essential to predict and prevent future emerging diseases. Here, we identify a high incidence of multiple paramyxoviruses in urine samples collected from a closed captive colony of circa 115 straw-coloured fruit bats (*Eidolon helvum*). The sequences detected have high nucleotide identities with those derived from free ranging African fruit bats and form phylogenetic clusters with the *Henipavirus* genus, *Pararubulavirus* genus and other unclassified paramyxoviruses. As this colony had been closed for 5 years prior to this study, these results indicate that within-host paramyxoviral persistence underlies the role of bats as reservoirs of these viruses.

## 1. Introduction

Most emerging infectious diseases presenting threats to public health are zoonoses originating in wildlife [1]. Understanding the ecology of zoonotic, or potentially zoonotic, viruses in their natural hosts is essential to predict and prevent disease emergence [2]. Natural hosts of an infectious agent are those that have co-evolved with the pathogen and are infected in nature without human intervention [3]. Several studies have identified bats as the most likely natural hosts for multiple emerging zoonotic viruses with high case fatality rates, such as SARS-like coronaviruses, Ebola and Marburg filoviruses, lyssaviruses and a range of paramyxoviruses (PVs) [3,4,5]. It has been hypothesised that bats share unique traits that enhance their potential as viral reservoirs such as long lifespans, large population sizes, high spatial mobility and high sympatry, which provides opportunities for pathogen persistence and for intra- and interspecific transmission of infectious agents [6,7]. 

Within the Paramyxoviridae, viruses in the Orthoparamyxovirinae and Rubulavirinae subfamilies have been detected in bats, and some of these have been associated with serious emerging zoonotic diseases [8]. Hendra and Nipah viruses (genus *Henipavirus*) were first detected in the 1990s after severe outbreaks of disease in domestic animals and humans. Hendra virus (HeV) was identified in Australia in 1994 causing fatal pneumonia and encephalitis in horses and humans [8]. Several variants of HeV were found to be widespread in Australian fruit bats (*Pteropus* spp.) and outbreaks of disease have occurred almost every year since its identification [9]. Nipah virus (NiV) emerged in Malaysia in 1998 causing fatal respiratory disease and encephalitis in pigs and humans [10]. Similarly, NiV was later detected in fruit bats (*Pteropus* spp.) in Malaysia and, subsequently, from pteropid bats elsewhere in South and Southeast Asia. Nipah virus continues to cause recurrent disease outbreaks in Bangladesh with proved bat-to-human and human-to-human transmission [11,12]. Therefore, fruit bats of the genus *Pteropus* are widely recognised as the natural reservoirs for HeV and NiV and their distribution was assumed to limit the range of henipaviruses [13].

Following HeV and NiV emergence, enhanced surveillance for potential pathogens associated with bats led to the discovery of a greater diversity of PVs in these taxa [14,15,16,17,18,19]. In 2008, henipavirus antibodies were found in the straw-coloured fruit bat (*Eidolon helvum*), a species of pteropodid bat, in Ghana without sympatry with the suspected Pteropus reservoirs [20]. Subsequent studies have detected several henipa-like virus sequences in faeces, urine and tissues of *E. helvum* and other African fruit bats [14,15,18]. Although the isolation and culture of African henipaviruses have not yet been achieved, a full genome sequence was obtained from *E. helvum* in Ghana, confirming its classification within the genus *Henipavirus* [14,21,22]. Despite evidence of African henipavirus spillover into humans [23], the potential to cause disease is unclear.

In addition, bat PVs in the subfamily *Rubulavirinae* have been recently discovered with unknown consequences for human or animal health. Menangle virus was isolated in Australia after an outbreak of reproductive disease in pigs, being later detected in humans with a febrile illness and in apparently healthy *Pteropus* sp. fruit bats [24]. Tioman virus was identified in Malaysian fruit bats with evidence of spillover to humans [25]. Furthermore, in 2008, three pararubulaviruses, or Tuhoko viruses, were isolated from fruit bat faeces in China [26], while three pararubulaviruses, Achimota pararubulavirus 1, 2 and 3 (AchPV1, AchPV2 and AchPV3), have been recently described in an urban population of *E. helvum* in Ghana [19,27]. There is widespread seropositivity to AchPV1 and to AchPV2 in *E. helvum* across sub-Saharan Africa, while antibodies to AchPV2 have been detected in people in Ghana [19]. While the clinical implications of zoonotic infection are unknown, experimental infections of ferrets with AchPV1 and AchPV2 were associated with respiratory disease [28].

*Eidolon helvum* is widely distributed throughout sub-Saharan Africa and it is the bat species most hunted for bushmeat in West Africa [29]. This species frequently forms large colonies in urban areas such as one that roosts in central Accra, Ghana containing up to one million individuals. The circulation of multiple PVs, including henipaviruses and pararubulaviruses, has been confirmed in this large urban population [14,15,19,30]. In January 2010, bats caught from the Accra colony were used to establish a closed captive colony of *E. helvum* in the vicinity of Accra Zoo with mesh double-walls, ground-level cladding and a solid roof to preclude direct or indirect contact with wild bats or other animals [31]. The captive colony was established with 77 bats of mixed age and sex and has been breeding successfully ever since. By April 2015, the colony had reached approximately 115 individuals through births only. Serological studies using a multiplexed microsphere assay demonstrated the presence of antibodies against henipaviruses in the colony over 24 months since the colony was closed, with evidence of maternal antibodies and later seroconversion in juveniles [31]. Additionally, using a NiV antibody binding assay, antibodies were detected from 2009 to 2017 with an average seroprevalence of 60.5% [32,33]. Longitudinal seroepidemiology shows indirect evidence for PV transmission within the colony [33], but virus detection had not been attempted to date. In this study, we aim to detect and characterise the PVs circulating in the *E. helvum* captive colony, using under-roost urine samples. The molecular examination of urine samples has been shown to be an effective way of detecting a range of PVs, including henipaviruses, excreted by fruit bats, including *E. helvum* [15,34].

## 2. Materials and Methods

Urine pools (*n* = 128) were collected from the closed captive colony of *E. helvum* in Accra Zoo over 23 time points from April to July 2015. Plastic sheets were suspended beneath the roosting area inside the bat enclosure (Figure 1) before feeding time, circa 4.00 p.m. Urine pools were collected the following morning circa 6.30 a.m. Urine pools visibly contaminated with faecal or plant material or diluted after heavy rain were not collected, but faecal contamination of samples could not be ruled out. A urine sample comprised of 500 μL of the urine pool preserved in 500 μL RNAlater (Invitrogen, Waltham, MA, USA). Urine samples were stored at −80 °C within 3 h of collection. At the end of the sampling period the samples were imported to the UK under permit in a cryogenic dry shipper (−146 °C) and the samples were stored at −80 °C until analysis.

Urine samples were vortexed for 30 s and centrifuged at 4000× *g* for 10 min to obtain a cell-free supernatant. RNA was extracted from 400 μL of supernatant using the MagMAX Viral RNA isolation kit (Applied Biosystems, Waltham, MA, USA) following the manufacturer’s protocol with carrier RNA replaced with linear polyacrylamide (Invitrogen, Waltham, MA, USA). All samples were further treated using the TURBO DNA-free Kit (Ambion, Austin, TX, USA) following the manufacturer’s instructions.

All samples were screened using pan-paramyxovirus hemi-nested RT-PCR (PAR-PCR) and Respirovirus-Morbillivirus-Henipavirus subgroup (RMH-PCR) using previously published primer sets [35] and modified PCR mixtures and thermocycling conditions. For the first PCR in the hemi-nested assay, we used the SuperScript III One-Step reverse transcription-PCR (RT-PCR) kit (Invitrogen, Waltham, MA, USA). The PCR mixture contained 1× reaction mix, 25 pmol of forward and reverse primers each, 10 nmol MgSO4, 1 μL of Superscript III RT/Platinum Taq mix and 2 μL aliquot of the RNA extract. Water was then added to achieve a final reaction volume of 25 μL. The thermocycler settings for the first reaction were: 60 °C for 1 min, 48 °C for 30 min, 94 °C for 2 min, 40× PCR cycles (94 °C for 2 min, 49 °C for 15 s, 68 °C for 1 min), 68 °C for 5 min and 4 °C until the end. For the second PCR in the hemi-nested assay, we used the Roche Expand High Fidelity PCR system (Roche, Basel, Switzerland). The PCR mixture contained 1× reaction buffer 3, 25 pmol of forward and reverse primers each, 50 nmol of MgCl_2_, 5 nmol of dNTP mix, 1.75 U of Expand High Fidelity Enzyme mix and 1 μL aliquot of the first reaction. Water was then added to achieve a final volume of 25 μL. The thermocycler settings for the second reaction were: 94 °C for 2 min, 40× PCR cycles (94 °C for 15 s, 49 °C for 30 s, 72 °C for 1 min), 72 °C for 5 min, 4 °C until the end. 

The PAR-PCR primer sets amplify a 530 bp fragment of highly conserved polymerase *L* genes. The RMH-PCR primer sets amplify a 439 bp fragment upstream of the polymerase *L* gene designed to be more specific for these PV genera.

PCR products were mixed with 5× DNA Loading Buffer (Qiagen, Hilden, Germany) and were electrophoresed in 2% (*w*/*v*) agarose gel followed by visualisation using GelGreen (Biotium, Fremont, CA, USA) nucleic acid stain and blue light. Positive bands of expected sizes were gel extracted using the MinElute kit (Qiagen, Hilden, Germany) and Sanger sequenced by a commercial laboratory (Eurofins Genomics, Ebersberg, Germany). 

Deduced viral sequences were aligned using Geneious software v11.1.5 [36]. Sequences were then run through the Basic Local Alignment Search Tool [37] to identify the similarity with previously published sequences (Appendix A).

For phylogenetic tree inference, publicly available sequences were downloaded from NCBI GenBank (Appendix A) and multiple alignments made using MUSCLE [38] in program MEGA X [39]. Maximum likelihood trees were constructed using model selection GTR+I+G [40] and 1000 bootstrap iterations in program MEGA X [39].

## 3. Results

PAR- and RMH-PCR assays produced seventeen and sixty-eight amplicons, respectively, from the urine samples tested (Table 1, Figure 2). Upon alignment, nine distinct sequences were identified (Appendix A), with close phylogenetic relationships between most sequences and with known PV sequences previously reported from *E. helvum* bats (Figure 3 and Figure 4). Three distinct sequences, Z15-U17P, Z15-U27P and Z15-U111P (Genbank MZ393370-2) were obtained using the PAR-PCR assay and six distinct sequences, Z15-12R, Z15-U17R, Z15-U27R, Z15-U78R, Z15-U86R and Z15-U115R (Genbank MZ393364-9) were obtained using the RMH-PCR assay.

The three sequences obtained using the PAR-PCR assay were located within three distinct clades throughout the *Paramyxoviridae* phylogeny (Figure 3). Sequences Z15-U17P and Z15-U27P showed high nucleotide identities (>95%) with previously detected sequences derived from *E. helvum* bats in Zambia [2] and Ghana [15], respectively. Sequence Z15-U17P grouped in the *Henipavirus* genus exhibiting 74.2% and 72.9% homology to NiV and HeV, respectively, whilst sequence Z15-U27P grouped with other henipa-related viral sequences. Sequence Z15-U111P clustered within a phylogenetically diverse subgroup of the *Pararubulavirus* genus that included AchPV1 and had 98% nucleotide identity with clones U69C/D [15] derived from Ghana.

Six distinct sequences were obtained with the RMH-PCR assay and these were scattered throughout the phylogenetic tree (Figure 4). Sequence Z15-U17R was located within the *Henipavirus* clade, while the other sequences were in diverse subgroups phylogenetically related to the henipaviruses. All sequences showed high nucleotide identities (>95%) with previously published sequences derived from *E. helvum* bats in Ghana, Gabon or the Republic of Congo [14,15,41], except for Z15-U115R, which is novel, having only 76% similarity to the nearest known sequences, those of clones (U58B and U32A) derived from *E. helvum* in Ghana [15].

Six of the nine sequences were detected during more than one sampling time point (Figure 2), with sequence Z15-U12R being the most frequently detected, in 13 out of 23 time points. Multiple sequences were most frequently observed during the mid-June time points.

## 4. Discussion

In this study, we detected the presence of multiple paramyxoviruses in our closed captive colony of *E. helvum* bats. The colony had been closed to new arrivals (other than births) for five years prior to the current study and the enclosure had double-mesh walls and a solid roof to prevent contact with bats or other free-living wildlife. To the best of our knowledge, this is the first report of persistence of multiple members of the *Paramyxoviridae* family within a small (~115 individuals), isolated bat population.

We found a range of PVs in our captive population of *E. helvum.* Analysis of the *L* gene fragments shows that the three and six distinct PV sequences detected by the PAR-PCR and RMH-PCR assays, respectively, belong to phylogenetically distinct subgroups. The six distinct and diverse sequences detected by the RMH-PCR and the pararubulavirus sequence detected using the PAR-PCR suggests that at least one henipavirus, one pararubulavirus and five unclassified PVs were simultaneously circulating in the colony. Six of nine sequences from the PAR- and RMH- PCR assays were almost identical (>95% homology) to sequences derived from the urban population of *E. helvum* in Accra, from which the captive colony originated [14,15,18]. These results, along with the isolation of the captive colony, suggest that the captive bats naturally acquired the infection in the original free-ranging population and the viruses were subsequently maintained in captivity over the following 5-year period. Only a small number of people are authorised to enter the bat cage, which is otherwise locked shut. This, along with the wearing of new or disinfected personal protective equipment, including coveralls and footwear, when entering the bat cage and the absence of any other bats being held in Accra Zoo reduces the likelihood of virus incursion via fomites. Although the cage construction precludes direct contact between wild and captive bats, and makes droplet transmission extremely unlikely, it cannot completely remove any possibility of aerosol transmission. While it is, therefore, possible that infection occurred subsequent to the founding of the captive colony via aerosol from wild bats or via fomites, it seems unlikely that this would be the case for such a large number of PVs. The sequences Z15-U17P, Z15-U17R, Z15-U78R and Z15-86R show great similarity to *E. helvum* derived sequences from Zambia, Gabon or the Republic of Congo [2,14,41]. Again, the founder bats were likely infected with these viruses at the time of capture, reinforcing previous observations of infection homogeneity across the panmictic *E. helvum* populations in Africa [30].

Six of the nine sequences were detected more than once during the course of the study. Some sequences were more frequently detected than others and the highest diversity of sequences was detected in the mid-June period. However, we are unable to comment on specific viral shedding patterns due to limitations in the methodology. The study was not conducted throughout the year and the sampling strategy to avoid collecting from urine pools contaminated with faeces or diluted with rainwater may have introduced a sampling bias, potentially missing urine from bats excreting virus. In addition, we did not test for the presence of PCR inhibition which may have affected virus detection. Nevertheless, the synchronous shedding of diverse PVs in pooled urine samples was detected using the same PCR assay. This could represent co-infection among individuals or could derive from multiple infected bats contributing to the same sample pool. Longitudinal studies on virus shedding in the colony and of the shedding status of individual bats are required to further determine the infection dynamics of PVs in this species.

Due to the differences in phylogenetic trees depending on the gene/fragment amplified, it is not possible to establish relationships between sequences derived from different assays. Sequences Z15-U17P amplified using the PAR-PCR and Z15-U17R amplified by the RMH-PCR may represent different fragments from the same or different viruses. The two sequences detected in the same urine pool may be explained by one or more bats excreting two viruses. However, as both sequences phylogenetically cluster closely with the *Henipavirus* clade, it is also possible that they belong to a single virus. Whilst they were both initially detected during the same time point in April 2015, Z15-U17R was detected during two further time points in May 2015 using the RMH-PCR assay. If both sequences do belong to the same virus, the absence of detection in the same urine pools using the PAR-PCR assay could be attributed to PCR inhibition or differences in assay sensitivity and specificity, as previously reported [15]. To confidently identify the viruses circulating in the captive colony, successful viral isolation and full genome sequencing are required.

In this study, we detected henipavirus-like sequences in multiple samples collected over a two-month period, showing repeated excretion over time. Conventional wisdom indicates that viruses such as PVs have a short infectious period and require large population sizes to enable persistence [42]. Patterns of PV infection in mammals other than bats suggest that minimum group size, usually above 100,000 individuals, is needed for pathogen persistence in a population [42,43]. In contrast, evidence of PV persistence in our closed captive colony of *E. helvum*, with approximately 115 individuals, suggests that at least some PVs can persist in small, isolated populations. This is consistent with longitudinal serological monitoring of this captive colony, whereby patterns of seroconversion in captive-born bats had led us to hypothesise that PVs were persisting and circulating within the colony over 10 years [33]. Similarly, antibodies against at least one henipavirus have been detected in an isolated, free-ranging population of *E. helvum* comprising <2500 individuals on Annobón island [44]. Taken together, these findings infer that at least some bat PVs might persist within individuals with long-term continuous or intermittent excretion. Persistent latent infection with recrudescence during periods of immune suppression has been suggested for NiV in *Pteropus* spp. [45] and observations of apparent increased henipavirus excretion, as inferred by increased antibody detection during periods of breeding or nutritional stress in fruit bats, may be consistent with this hypothesis [31,46]. Recent models of PV transmission dynamics in bat populations using age-specific serological data indicate that population-level persistence could be reliant on within-host persistence [47]. Longitudinal paired serology–virology data and studies on bat immune responses to PV infection are required to further investigate mechanisms of PV persistence in bat populations.

Phylogenetic analyses have placed bats as tentative ancestral hosts of major mammalian PVs [14]. Additionally, sequencing of highly conserved motifs in the PV genome has shown that African rather than Asian henipavirus clades are identical to the viral ancestors [14]. Therefore, it is suggested that all henipaviruses, including HeV and NiV, have evolved from a common ancestor of African origin [14]. The diversity of *L* gene sequences, including a likely novel sequence (Z15-U115R), detected within our small captive population further support the hypothesis that bats are the natural reservoirs for PVs [14].

These findings suggest that additional diversity of PVs with zoonotic potential exists in bat populations, particularly in Africa, and the study of these viruses might help to inform the detection of future spillover events. However, increased efforts are needed to fully characterise PV diversity in African bats in order to elucidate viral infection dynamics and mechanisms of persistence in bat populations. The absence of any observed disease outbreaks or unusual mortality events in the captive bat colony supports findings from Ghana and elsewhere that PV infection does not cause clinically obvious ill health in fruit bats, e.g., [15,17,18,19]. It is possible, however, that infection with one or more of the PVs detected does have a more nuanced negative impact on individual fitness. Detailed histopathological and virological investigations of freshly dead bats and longitudinal studies of bats naturally or experimentally infected with different PVs are required to provide a better understanding of the impacts of infection.

Our unique opportunity to study a small colony of captive *E. helvum* has found that multiple paramyxoviruses can persist over at least a 5-year period. To our knowledge, we provide the first evidence of viral persistence in an isolated population of fewer than 150 individuals. Phylogenetic analyses of detected viral sequences show that a great diversity of paramyxoviruses was present in the colony, including one potentially novel viral sequence. Though we are not able to confidently report on viral transmission or dynamics, further studies to fully characterise these paramyxoviruses and longitudinal studies would improve our understanding of these. Close relationships between some sequences and known human pathogens in the *Henipavirus* and *Pararubulavirus* genera circulating in free-living *E. helvum* was observed. *Eidolon helvum* forms large roosts in both rural and urban areas across much of Africa and is widely hunted [29,48], presenting multiple opportunities for human exposure to bat paramyxoviruses.

## Figures and Tables

**Figure 1 viruses-13-01659-f001:**
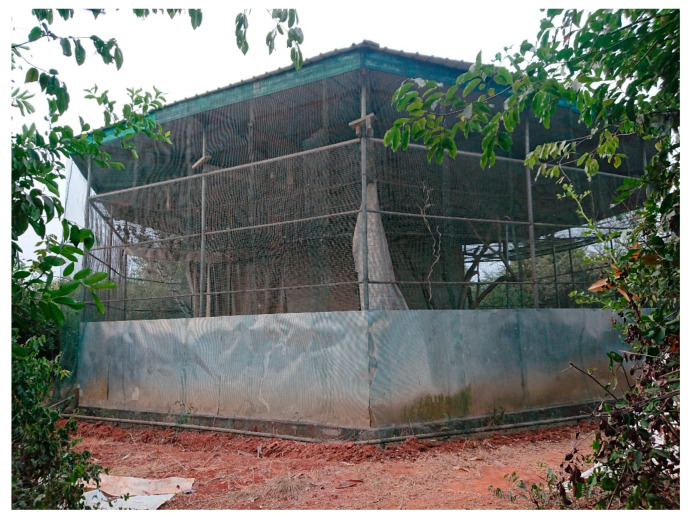
Cage housing the *Eidolon helvum* bat colony. The mesh double-walls, ground-level cladding and outer solid roof are clearly visible. Note, the bat colony can be seen roosting while hanging from the inner mesh roof towards the left side of this photograph.

**Figure 2 viruses-13-01659-f002:**
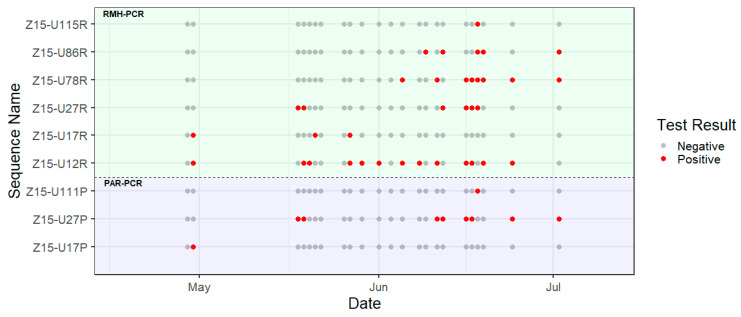
Timeline of amplicon sequences detected using the PAR-PCR (lower panel) and RMH-PCR assays (upper panel). The red dot denotes the positive amplicon sequence and the grey dot denotes the negative amplicon sequence detected at each time point.

**Figure 3 viruses-13-01659-f003:**
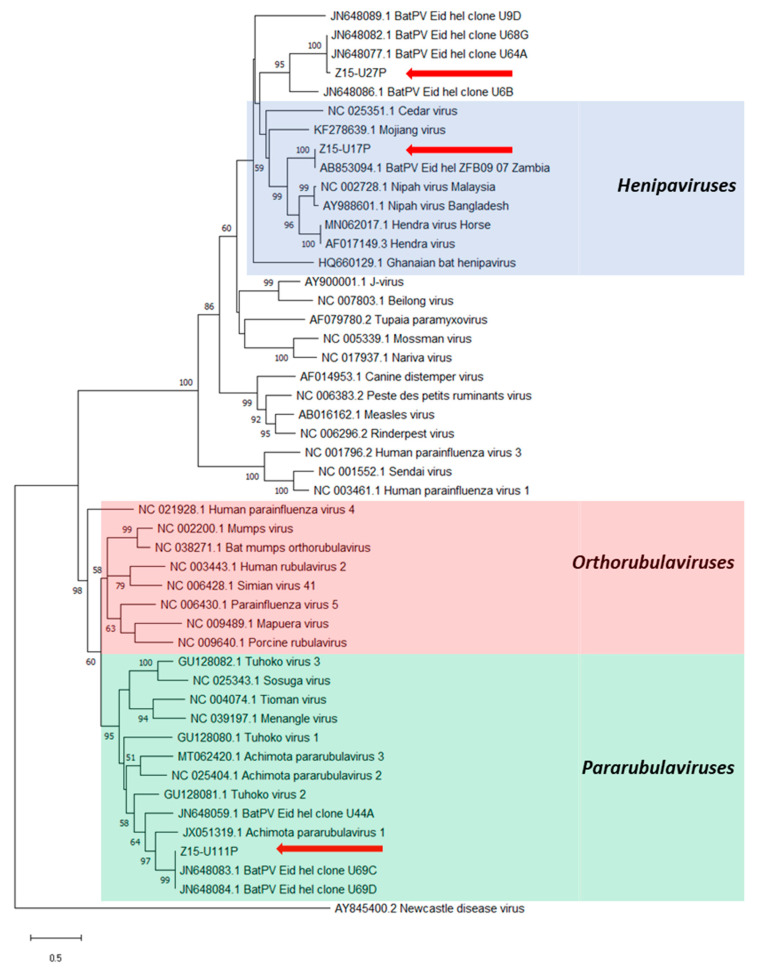
Phylogenetic analysis of partial *L* gene sequences obtained after PAR-PCR on *E. helvum* urine samples (red arrows). Maximum likelihood tree with bootstrapping (1000 iterations) generated in MEGA X, using 530 bp alignment against publicly available paramyxovirus sequences (NCBI Genbank) and outgroup Newcastle disease virus. Bootstrap values for 1000 replicates are indicated as percentages (where >50%) and the number of nucleotide substitutions per site is to scale as indicated by the scale bar.

**Figure 4 viruses-13-01659-f004:**
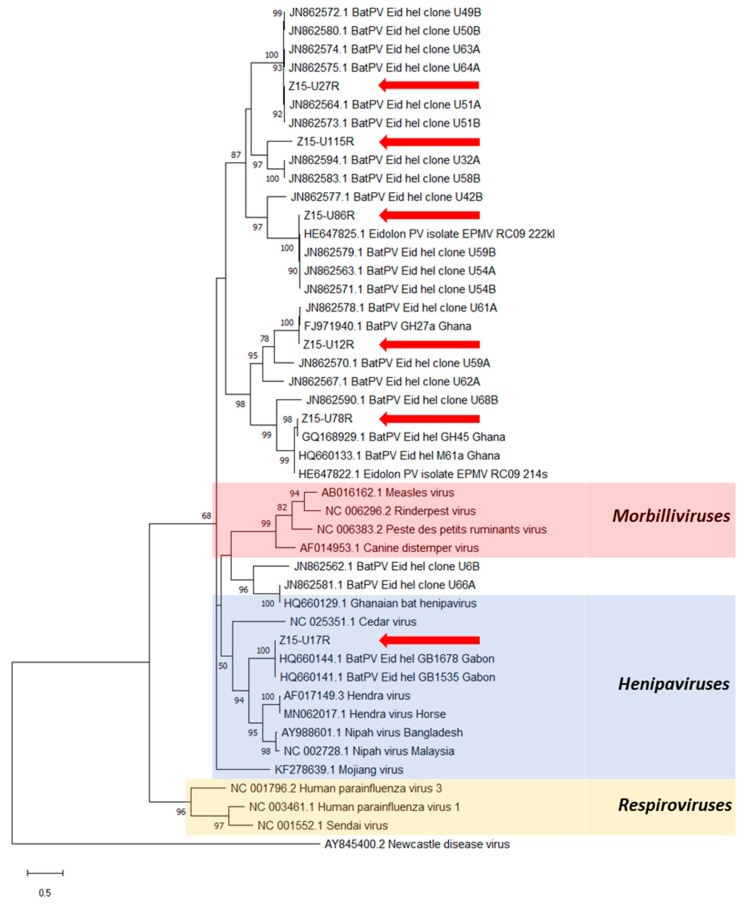
Phylogenetic analysis of partial *L* gene sequences obtained after RMH-PCR on *E. helvum* urine samples (red arrows). Maximum likelihood tree with bootstrapping (1000 iterations) generated in MEGA X, using 439 bp alignment against publicly available paramyxovirus sequences (NCBI Genbank) and outgroup Newcastle disease virus. Bootstrap values for 1000 replicates are indicated as percentages (where >50%) and the number of nucleotide substitutions per site is to scale as indicated by the scale bar.

**Table 1 viruses-13-01659-t001:** The number of urine samples tested at each sampling time point and corresponding number of positive amplicons detected for PAR- and RMH-PCR assays.

Date	No. of Samples Tested	No. of PCR Positive Amplicons	
		PAR-PCR	RMH-PCR
29/04/2015	10	0	0
30/04/2015	15	1	7
18/05/2015	5	4	4
19/05/2015	5	1	3
20/05/2015	5	0	1
21/05/2015	5	0	1
22/05/2015	5	0	0
26/05/2015	3	0	0
27/05/2015	5	0	4
29/05/2015	5	0	1
01/06/2015	5	0	3
03/06/2015	5	0	0
05/06/2015	5	0	4
08/06/2015	5	0	5
09/06/2015	5	0	4
11/06/2015	5	1	2
12/06/2015	5	1	2
16/06/2015	5	4	4
17/06/2015	5	2	5
18/06/2015	5	1	5
19/06/2015	5	0	5
24/06/2015	5	1	4
02/07/2015	5	1	4

## Data Availability

Data is contained within the article or Appendix A.

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
