# Peer review of "Persistence of Multiple Paramyxoviruses in a Closed Captive Colony of Fruit Bats (Eidolon helvum)"

_viruses, 2021, doi:10.3390/v13081659_

Round 1

Reviewer 1 Report

I think the manuscript reads well and have no specific comments to the content. However, there are some minor typing errors:

  • Line 3: Eidolon helvum should be italicized.
  • Line 33:  circa. should be written without . or replaced by approximately as used later in the manuscript.
  • Lines 52: One p is lacking in sp. or the latin name may be omitted as it is mentioned in line 49.
  • Line 272: One p is lacking in sp.
  • Line 300: Eidolon can be shortened to E. for consistency.

Author Response

I think the manuscript reads well and have no specific comments to the content. However, there are some minor typing errors:
We thank the reviewer for this observation.

Line 3: Eidolon helvum should be italicized.

– This revision has now been made.

Line 33: circa. should be written without . or replaced by approximately as used later in the manuscript.

- This revision has now been made.

Lines 52: One p is lacking in sp. or the latin name may be omitted as it is mentioned in line 49.

- This revision has now been made.

Line 272: One p is lacking in sp.

This revision has now been made.

Line 300: Eidolon can be shortened to E. for consistency.

– This revision has not been made as it is grammatically incorrect to start a sentence with an abbreviation.

Reviewer 2 Report

Gibson et al. sought to identify whether Paramyxovirus RNA could be detected in a colony of Eidolon helvum bats maintained in relative isolation from other bats, and showing serological evidence of ongoing Paramyxovirus infection. Urine from the colony was collected for a period two years after the colony was closed off from close contact with external sources, and a number of Paramyxovirus sequences were detected by RT-PCR and nested PCR using two sets of standardized Paramyxovirus-specific primers for the highly-conserved polymerase gene. The authors acknowledge the limitations of the study (different sets of primers may be identifying the same virus, study was not longitudinal enough to gain seasonal or other epidemiological information, and pooled urine poses challenges for sensitivity).

Comments: This was a very clear, transparently-reported study with implications for our understanding of Paramyxovirus persistence and pathology in bats. I hope the authors are pursuing full-length sequences for PVs in the samples with positive signal; optimally, if there are stored blood/serum samples from this time frame, they should also be pursued. I also hope the authors will assay these samples with the AR primers for Rubulavirus/Orthorubulavirus detection in the future. Finally, as the bats age, are corpses collected for assesment of organs for “latent”/persistent infection? The authors appear to suggest that all detected viruses must be present in the colony from the beginning of its isolation, and so detection of bats harbouring such “silent” infection would be a strong supporting piece of information. Of course, these questions are outside the scope of the current work, but I look forward to future publications addressing them. Finally, I was going to recommend that the researchers look in to capacity-building collaborations with the team maintaining this bat colony and with other research opportunities in Low/Middle-income countries (see: https://doi.org/10.1371/journal.pmed.1001612 for context), but their second submission for this series indicates that they have already done so.

Minor Revisions:

In Supplementary Table S2, and Figs. 2 and 3, “Ghana virus” should be renamed to “Ghanaian bat henipavirus” per current ICTV convention (see: https://talk.ictvonline.org/taxonomy/p/taxonomy-history?taxnode_id=202001608). Using human-habited place-names to designate viruses is frowned upon.

I suggest the authors comment on the implications for our understanding of the pathogenic potential of these viruses in bats. Conventional wisdom is that these viruses are not (typically) significantly pathological in bats, but studies of wild-caught animals cannot rule out that a percentage of animals are severely sickened and/or die from infection with a given PV without detection by researchers. This colony currently supports the former hypothesis.

The authors should comment on the likelihood of transmission of outside viruses to the colony. The barriers described appear to prevent direct physical contact, but is there reasonable possibility for aerosol transmission from respiratory or urinary secretions by outside bats? What precautions are taken by the staff that maintain the colony for preventing tracking outside urine/secretions into the colony, and is such a transmission route likely? A discussion of how to evaluate the likelihood of these events would be helpful; I believe there is limited literature on this, and so the manuscript would benefit from identifying the holes in our current understanding of these viruses and their transmission routes and infectivities. The authors should also consider including an illustrative figure of the colony setup, similar to Figure 1 in their sister submission.

Author Response

Comments: This was a very clear, transparently-reported study with implications for our understanding of Paramyxovirus persistence and pathology in bats. I hope the authors are pursuing full-length sequences for PVs in the samples with positive signal; optimally, if there are stored blood/serum samples from this time frame, they should also be pursued. I also hope the authors will assay these samples with the AR primers for Rubulavirus/Orthorubulavirus detection in the future. Finally, as the bats age, are corpses collected for assesment of organs for “latent”/persistent infection? The authors appear to suggest that all detected viruses must be present in the colony from the beginning of its isolation, and so detection of bats harbouring such “silent” infection would be a strong supporting piece of information. Of course, these questions are outside the scope of the current work, but I look forward to future publications addressing them. Finally, I was going to recommend that the researchers look in to capacity-building collaborations with the team maintaining this bat colony and with other research opportunities in Low/Middle-income countries (see: https://doi.org/10.1371/journal.pmed.1001612 for context), but their second submission for this series indicates that they have already done so.

We thank the reviewer for these comments and for their interest in our study. As the reviewer suggests, we are currently attempting to conduct whole genome sequencing of the identified viruses. The pathological and virological investigation of bat carcases following natural death in the colony is a good one and one we have also been trying to pursue, but it is amazing how difficult it is to obtain suitably fresh carcases: the colony is free-flying in a very large cage with a natural vegetation understorey and the heat can be intense in Accra. We will keep trying.

Minor Revisions:
In Supplementary Table S2, and Figs. 2 and 3, “Ghana virus” should be renamed to “Ghanaian bat henipavirus” per current ICTV convention (see: https://talk.ictvonline.org/taxonomy/p/taxonomy-history?taxnode_id=202001608). Using human-habited place-names to designate viruses is frowned upon.

- This revision has now been made.

I suggest the authors comment on the implications for our understanding of the pathogenic potential of these viruses in bats. Conventional wisdom is that these viruses are not (typically) significantly pathological in bats, but studies of wild-caught animals cannot rule out that a percentage of animals are severely sickened and/or die from infection with a given PV without detection by researchers. This colony currently supports the former hypothesis.

- We have now addressed this issue in the Discussion, lines 318-325.

The authors should comment on the likelihood of transmission of outside viruses to the colony. The barriers described appear to prevent direct physical contact, but is there reasonable possibility for aerosol transmission from respiratory or urinary secretions by outside bats? What precautions are taken by the staff that maintain the colony for preventing tracking outside urine/secretions into the colony, and is such a transmission route likely? A discussion of how to evaluate the likelihood of these events would be helpful; I believe there is limited literature on this, and so the manuscript would benefit from identifying the holes in our current understanding of these viruses and their transmission routes and infectivities.

- We have now addressed the issue of biosecurity and the likelihood of transmission from free-living wild bats to the captive bats in the Discussion, lines 244-253. With the paucity of data on routes of transmission and infectious doses, however, we felt it too speculative to elaborate any further on this matter. Hopefully, this paper will help inform further studies in this area.

The authors should also consider including an illustrative figure of the colony setup, similar to Figure 1 in their sister submission.
- We have inserted a new figure (now Figure 1) to illustrate the colony set-up. The other figures remain and have been renumbered accordingly. In addition, a scale bar has been added to each phylogenetic tree, as requested by Reviewer 2 elsewhere.